https://doi.org/10.5194/hess-2024-71





# A novel framework for accurately quantifying wetland
# depression water storage capacity with coarse-resolution
# terrain data
Boting Hu[1,2], Liwen Chen[1], Yanfeng Wu[1], Jingxuan Sun[1,2], Y. Jun Xu[3], Qingsong
Zhang[1,2], Guangxin Zhang[1]
[1] Northeast Institute of Geography and Agroecology, Chinese Academy of Sciences, Changchun, Jilin
130102, China
[2] University of Chinese Academy of Sciences, Beijing 100049, China
[3] School of Renewable Natural Resources, Louisiana State University Agricultural Center, 227
Highland Road, Baton Rouge, LA 70803, USA
*Correspondence to*: Guangxin Zhang (zhgx@iga.ac.cn)
**Abstract.** Accurate quantification of wetland depression water storage capacity (WDWSC) is
imperative for comprehending the wetland hydrological regulation functions to support integrated
water resources management. Considering the challenges posed by the high acquisition cost of
high-resolution LiDAR DEM or the absence of field measurements for most wetland areas, urgent
attention is required to develop an accurate estimation framework for WDWSC using open-source,
low-cost, multi-source remote sensing data. In response, we developed a novel framework,
WetlandSCB, utilizing coarse-resolution terrain data for accurate estimation of WDWSC. This
framework overcame several technical difficulties, including biases in above-water topography,
incompleteness and inaccuracy of wetland depression identification, and the absence of bathymetry.
Validation and application of the framework were conducted in two national nature reserves of
northeast China. The study demonstrated that integrating priority-flood algorithm, morphological
operators and prior information can accurately delineate the wetland depression distribution with
overall accuracy and Kappa coefficient both exceeding 0.95. The use of water occurrence map can
effectively correct numerical biases in above-water topography with Pearson coefficient and $R^2$
increasing by 0.33 and 0.38 respectively. Coupling spatial prediction and modeling with remote sensing
techniques yielded highly accurate bathymetry estimates, with <3% relative error compared to filed
measurements. Overall, the WetlandSCB framework achieved estimation of WDWSC with <10%





relative error compared to field topographic and bathymetric measurements. The framework and its
concept are transferable to other wetland areas globally where field measurements and/or
high-resolution terrain data are unavailable, contributing to a major technical advancement in
estimating WDWSC in river basins.
**Keywords**: Wetland depression; Water storage capacity; Hypsometric curve; coarse-resolution
terrain data; wetland hydrological regulation functions

**1  Introduction**
Wetlands are multifunctional ecosystems considered as nature-based solutions for effective water
management in river basins (Thorslund et al., 2017). They exert a profound influence on watershed
hydrological processes and water resource availability through their hydrological regulation functions,
such as maintaining baseflow, buffering floods, and delaying droughts (Acreman and Holden, 2013;
Wu et al., 2023). These functions are essential for enhancing watershed resilience and ensuring water
security (Cohen et al., 2016; Evenson et al., 2018; Lane et al., 2018). Wetland depression water storage
capacity (hereafter abbreviated as WDWSC) represents a critical component of wetland hydrological
regulation functions. The quantitative study of the WDWSC to advance scientific insights into wetland
hydrological regulation functions and support integrated water resources management (Ahmad et al.,
2020; Fang et al., 2019; Jones et al., 2018; Shook et al., 2021).
The WDWSC can be defined as the maximum surface water volume that each wetland depression
can store without spilling to down-gradient waters (Jones et al., 2018). Previous studies predominantly
employed wetland depression identification algorithms to derive wetland depression topography from
terrain data. Subsequently, hypsometric curves (area-depth) are constructed based on the derived
topography. Finally, the integration of the hypsometric curves is solved to determine the WDWSC (e.g.,
Haag et al., 2005; Wu and Lane, 2016). Therefore, the key determinants for the accuracy of the
WDWSC calculation are the rationality of the wetland depression identification algorithms and the
precision of terrain data. Many scholars have conducted research on wetland depression identification
algorithms, which can be mainly categorized into three types: depression filling, depression breaching
and hybrid combing both the filling and breaching approaches. Among these, the priority-flood
algorithm within the depression filling category is widely adopted as a prevalent algorithm for wetland
depression identification (Barnes et al., 2014; Lindsay, 2016; Wu et al., 2019; Zhou et al., 2016). The





priority-flood algorithm works by flooding DEM cells inwards from their edges using a priority queue to determine the sequence of cells to be flooded. Wu et al. (2019) and Rajib et al. (2019) demonstrated the feasibility of accurately deriving wetland depression topography using the priority-flood algorithm in the Pipestem watershed and Upper Mississippi river basin, respectively. Bare-earth high-resolution airborne light detection and ranging (LiDAR) DEM can provide accurate topographic information of wetland depressions, significantly improving the estimation accuracy of the WDWSC. For example, Jones et al. (2018) used high-resolution LiDAR DEM to estimate WDWSC in the Delmarva Peninsula. However, the high acquisition cost of LiDAR DEM renders it impractical for large-scale estimation of WDWSC. The global open-access spaceborne-derived DEMs (hereafter referred as global DEMs), such as Shuttle Radar Topography Mission (SRTM), ALOS Global Digital Surface Model, the Terra Advanced Spaceborne Thermal Emission and Reflection Radiometer (ASTER) Global Digital Elevation Model, offer topographic information at a fine spatial scale. However, compared to the bare-earth LiDAR DEM, the global DEMs exhibit three obvious limitations. First, radar altimetry cannot penetrate water surfaces, so the global DEMs produced from radar altimetry do not provide any bathymetric information. Second, in certain regions, there may be substantial numerical discrepancies in above-water topography. Third, the global DEMs often suffer from lower horizontal and vertical resolutions. Due to the limitations in global DEMs, delineation of wetland depressional areas using the advanced priority-flood algorithm also suffers from three problems: the bias in above-water topography (Fig. 1a and 1b), incompleteness and inaccuracy of wetland depressions identification (Fig. 1c), and the absence of bathymetric information (Fig. 1d).



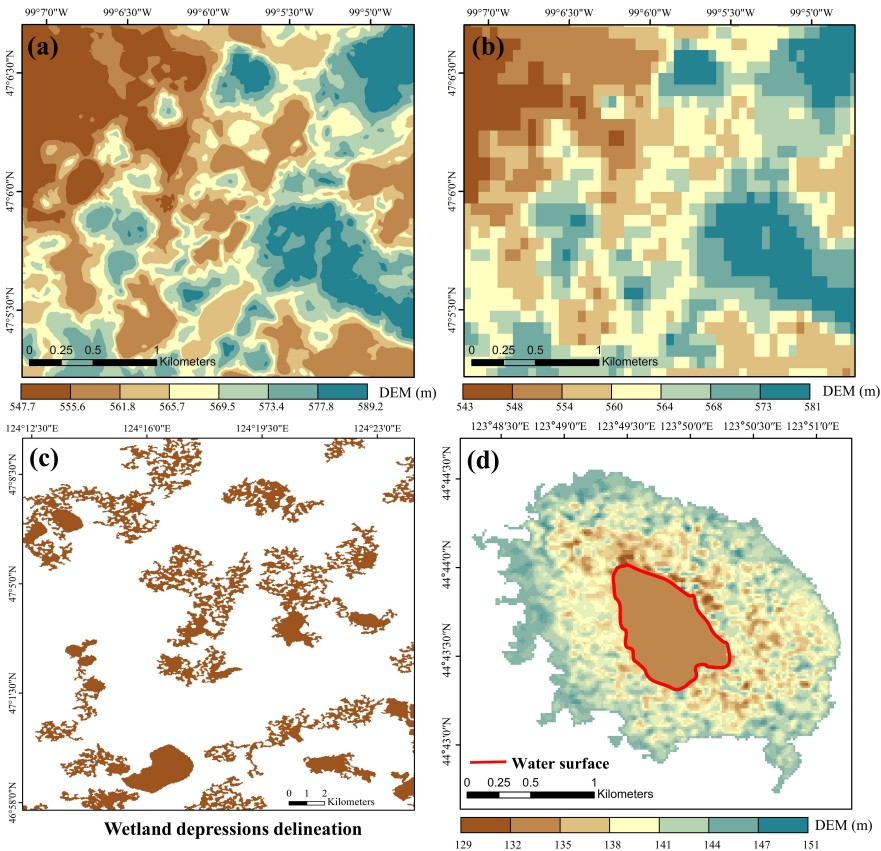

**Figure 1: Wetland depression extraction based on the priority-flood algorithm and global DEMs suffers from the bias of above-water topography (Figures 1a and 1b show the discrepancies in above-water topography obtained from LiDAR DEM and ALOS DEM, respectively, in the Prairie Pothole Region of North Dakota), incompleteness and inaccuracy of wetland depressions identification (Fig. 1c), and the absence of bathymetric information (Figure 1d, where the entire water surface is represented by a single elevation value of 129 m).**

In an effort to minimize the impact of the absence of bathymetric information in global DEMs on the estimation accuracy of the WDWSC, researchers have conducted studies on the estimation of underwater hypsometric relationship of wetland depressions, and the methods can be divided into two types: spatial prediction and modeling methods and remote sensing technologies. The spatial prediction and modeling methods assume that the bathymetry can be considered as a spatial extension of the





surrounding exposed terrains due to long-term tectonic and geophysical evolution processes.
Consequently, the underwater hypsometric relationship is assumed to be fundamentally similar to the
above-water hypsometric relationship in wetland depressions (e.g., Ahmad et al., 2020; Bonnema et al.,
2016; Bonnema and Hossain, 2017; Liu and Song, 2022; Tsai et al., 2010; Vanthof and Kelly, 2019;
Verones et al., 2013; Wu and Lane, 2016; Xiong et al., 2021). However, the large numerical bias in the
above-water topography of global DEMs in certain regions can distort the constructed above-water
hypsometric relationship of wetland depressions, thus introducing significant uncertainty to the
underwater hypsometric relationship estimated by this method. Over the past few decades, remote
sensing technologies have demonstrated remarkable capabilities in estimating underwater hypsometric
relationships at large spatial scales, facilitated by the rapid emergence of various advanced satellite
sensors, including optical, passive microwave, and radar instruments (Duan and Bastiaanssen, 2013;
Gao et al., 2015; Liu et al., 2022). The commonly employed approach for estimating underwater
hypsometric relationship requires simultaneous observations of water area provided by optical images
(e.g., Landsat series) and the corresponding water level provided by altimetry satellites (e.g., Sentinel-3,
CryoSat-2, Envisat). However, accuracy challenges arise due to numerical biases of altimetry satellites,
cloud contamination in some optical images, and the occasional occurrence of one water area value
corresponding to multiple water level values or vice versa (Li et al., 2019a; Liu et al., 2024). In
summary, previous studies have mainly utilized LiDAR DEM data to estimate WDWSC (e.g., Jones et
al., 2018; Huang et al., 2011; Kessler and Gupta, 2015; Land and D'Amico, 2010; Wu et al., 2016; Wu
et al., 2019). However, these studies have seriously overlooked the issues of incompleteness and
inaccuracy of wetland depression identification, as well as the bias in above-water topography,
resulting in a high level of uncertainty in the WDWSC estimation. In addition, insufficient attention has
been paid to the drawbacks and limitations of both spatial prediction and modeling methods and remote
sensing technologies in estimating bathymetry. Consequently, a comprehensive and systematic solution
for the accuracy estimation of WDWSC based on the global DEMs has not yet been developed.

Therefore, this study aims to develop a framework for accurately estimating WDWSC by

integrating multi-source remote sensing data and prior knowledge. Specifically, we integrated
priority-flood algorithm, morphological operators and prior information on water distribution map to
delineate the spatial extent of wetland depressional areas. We then corrected the bias in above-water
topography based on water occurrence map. Finally, we utilized remote sensing techniques to couple





spatial prediction and modeling to estimate bathymetry of wetland depressional areas. The principle
contribution of this developed framework, termed as WetlandSCB, lies in addressing the challenges
hindering the improvement of accuracy in estimating WDWSC based on global DEMs.
**2 Methodology**
The WetlandSCB framework can be summarized in four steps as illustrated in Figure 2. Step 1
delineation of wetland depressional areas; Step 2 above-water topography reconstruction; Step 3
bathymetric information estimation; and Step 4 hypsometric curve construction and WDWSC
calculation. Each of the four steps are described in the following sections.

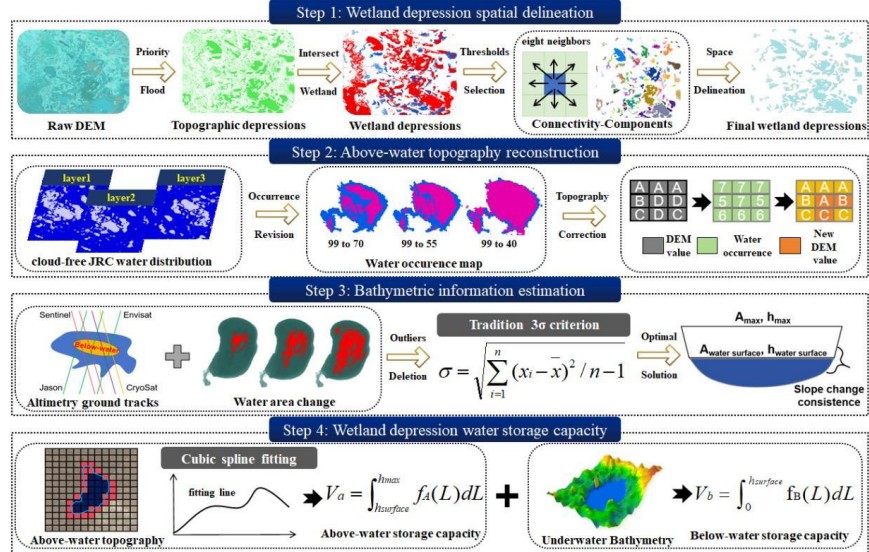


**Figure 2: Flowchart of the WetlandSCB framework for accurate estimation of wetland depression water**
**storage capacity (WDWSC) comprising four technical steps. In step 1, spatial distribution of wetland**
**depressional areas are delineated. In step 2, wetland above-water topography is reconstruction. In step 3,**
**bathymetric information of wetland depressional areas is estimated. In step 4, a hypsometric curve (i.e.**
**depth-area relation) is developed and WDWSC is quantified.**

**2.1 Wetland depression spatial delineation**
We extracted the original wetland depression map from the global DEMs based on the
priority-flood algorithm and wetland maps (Fig. 3). To eliminate the artifact wetland depressions, it
was necessary to transform the wetland depression map into a binary image consisting of pixels that





area labeled as logical ones (wetland depression) and zeros (non-wetland depression). We then
employed the eight-neighbor connectivity algorithm to extract the spatial extent of each wetland
depression from the binary image. Subsequently, the circularity (Eq. 1) and eccentricity (Eq. 2)
indicators were used to exclude the artifact wetland depressions (Ahmad et al., 2020) as follows:
$$Circularity = \frac{P}{2\sqrt{\pi \cdot A}} \qquad (1)$$
$$Eccentricity = \frac{D_f}{l_m} \qquad (2)$$
where $P$ (m) and $A$ (m²) are the perimeter and area of the wetland depression, respectively. $D_f$ (m)
and $L_m$ (m) represent distance between foci and the length of major axis of wetland depression.

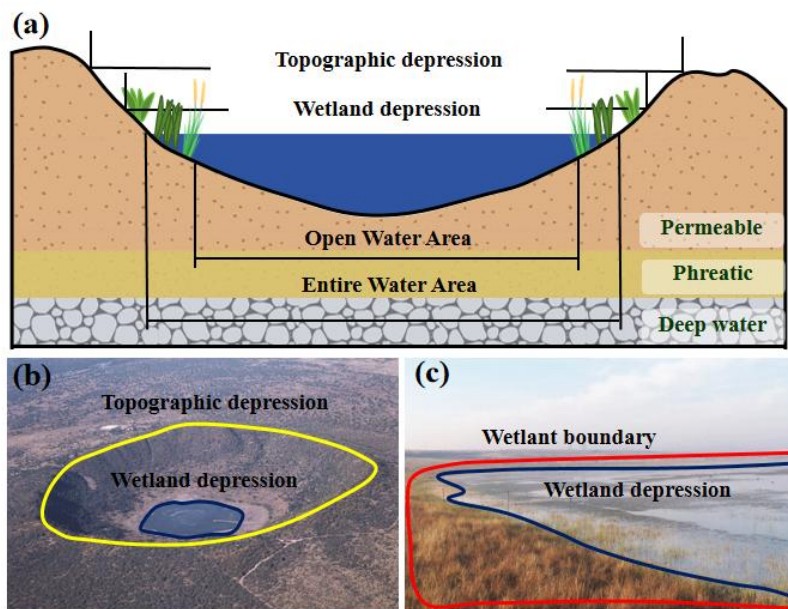


**Figure 3. (a) Conceptual diagram of wetland depression profile. (b) and (c) show the two representative**
**wetland depressional areas located in South Africa (modified from De Klerk et al., 2016).**

Due to incompleteness and inaccuracy identification of some wetland depressions in the original
wetland depression map (Figure 4a), morphological operators of erosion and dilation are applied for the
initial spatial processes (Figure 4b). The erosion operator erodes away the boundaries of wetland
depressions to enhance their edges and remove noise. The dilation operator fills up small holes
(non-wetland depression pixels) surrounded by a group of wetland depression pixels (Pulvirenti et al.,
2011a). The combined effect of the two operators is to remove noises while preserving the substantive
features in the image. The water distribution map, which serves as prior information, effectively
characterizes the spatial extent of wetland depressions (Figure 3). Therefore, the wetland depression
map, after being processed by the morphological operators, is then intersected with the water
distribution map to obtain a complete and final wetland depression map (Figure 4c).

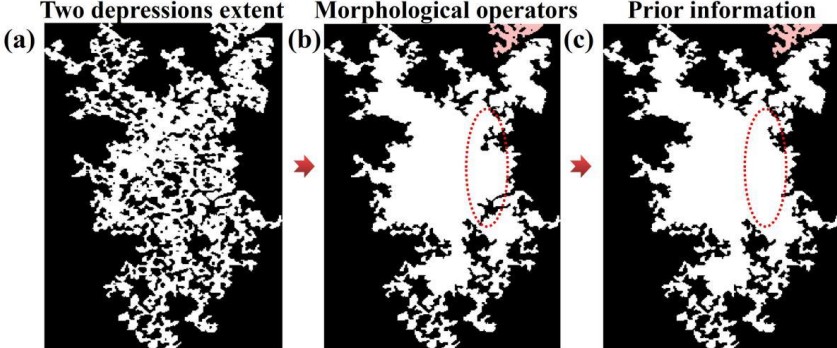

**Figure 4. The wetland depression map based on the morphological operators and priori information on the**
**water distribution map.**

**2.2    Above-water topography reconstruction**

The water occurrence map can effectively describe three-dimensional topography at a large spatial

scale (Armon et al., 2020; Li et al., 2019b). The water occurrence map is generated by summing the
times that the pixel is detected as water and dividing it by the number of total valid observations.
Therefore, if there is a accurate water occurrence maps, a close relationship between the water
occurrence and the topography for wetland depressions can be found (Li et al., 2021). The open-source
Global Surface Water Mapping Layers produced by the European Commission's Joint Research Centre
(JRC) contains a water occurrence map, which has been widely used to describe the topography of
wetland depressions globally or in different regions (Luo et al., 2019; Pickens et al., 2020; Yao et al.,
2019; Zou et al., 2018). However, due to the temporal discontinuity of cloud-free JRC water
distribution images, they are more available during dry seasons than wet seasons, leading to deviations
in the representation of real topography at the scale of individual wetland depression (Chu et al., 2020).

To address the above issue, this study proposes a method to restore the cloud-contaminated JRC





water distribution images to improve the accuracy of the JRC water occurrence map. For wetland
depressional areas, the JRC water distribution images are classified into cloud-free and
cloud-contaminated images using the cloud screening algorithm of the Google Earth Engine platform.
The Canny edge detection algorithm is used to obtain the water body boundary of the two types of
images. Theoretically, if the water areas are the same, the water body boundary of the cloud-free image
should overlap with the exposed water body boundary in the cloud-contaminated image (Figure 5a).
Therefore, by overlapping the water body boundaries of the cloud-free images with the
cloud-contaminated images, the missing spatial extent of water bodies in the cloud-contaminated
images can be filled.
The corrected JRC water occurrence map is utilized to reconstruct above-water topography. This
is because the water occurrence values within the same wetland depression correspond to elevation
values (Figure 5b and 5c). However, each corrected water occurrence value may correspond to multiple
elevation values in the global DEMs. Therefore, the median of multiple elevation values is used as the
unique elevation value corresponding to the water occurrence value.

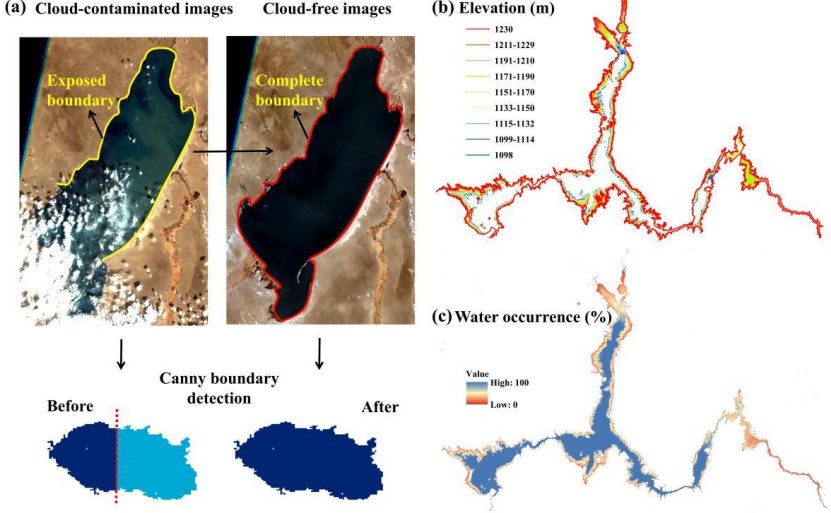


**Figure 5. Above-water topography reconstruction of wetland depressional areas. (a) Restoration method of**
**cloud-contaminated JRC water distribution images. (b) LiDAR DEM and JRC water occurrence map of**
**Mead Lakes in the United States.**

**2.3   Bathymetric information estimation**





The remote sensing technologies are used to estimate the underwater bathymetry of wetland
depressions, and the similarity between the underwater and above-water hypsometric relationships is
served as an evaluation criterion to seek for the optimal solution within the estimated results that
accurately represents underwater bathymetry based on the principle of spatial prediction and modeling
methods.
The outliers in the underwater are-level pairs are removed using the 3-sigma rule. As the slope
profile is a crucial indicator reflecting the hypsometric relationship of wetland depressions (Clark and
Shook, 2022; Sjöberg et al., 2022). Therefore, we first form various combinations of the processed
underwater area-level pairs (each water area value uniquely corresponds to a water level value in each
combination), and calculate the slope profile value $p_u$ for each combination. Then the combination with
$p_u$ closest to the above-water slope profile $p_a$ is taken as the optimal solution, which can effectively
represent underwater bathymetry of wetland depressions.
In this study, a logarithmic transformation is applied to the calculation formula for the slope
profile $p$ of wetland depressions established by Hayashi and Van der Kamp (2000) to obtain Eq. 3. The
least squares method is used to solve Eq. 3 to obtain the slope profile $p$ value of wetland depressions:
$$P = \frac{2 \cdot \ln(h_w / h_d)}{\ln(A_w / A_d)} \tag{3}$$
where $h$ (m), $A$ (m$^2$) represent the depth and area of wetland depressions, and $w$ and $d$ represent
the different area-depth pairs.

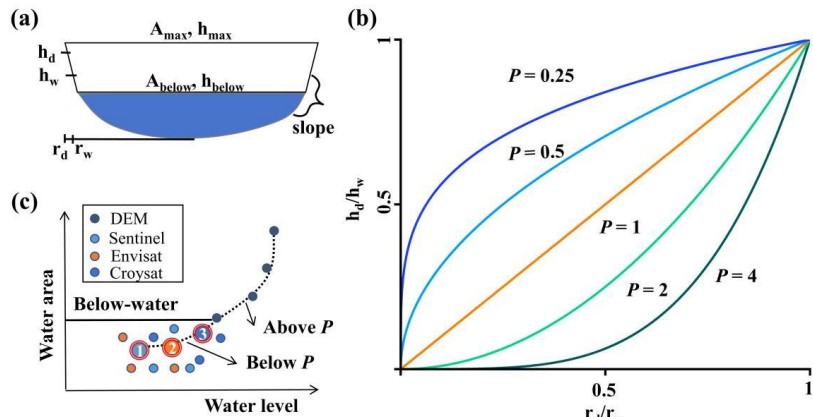


**Figure 6. Estimation of bathymetric information for wetland depressional areas. (a) Schematic**
**representation of a simplified wetland depression profile, where $h$ (m), $r$ (m) and $A$ (m$^2$) represent the depth**


of a wetland depressional area, the distance between the edge and the center of the wetland depression, and
the area of the wetland depression, respectively. (b) Wetland depression profile for various *p* values. (c)
Methods for bathymetric estimation of wetland depressions, where Sentinel, Envisat, and Croysat are
different altimetry satellites, and the numbers 1, 2, and 3 are selected depth-area pairs.

**2.4   Estimation of wetland depression water storage capacity**

Deriving the area-level hypsometric relationship from the corrected above-water topography and

estimated underwater bathymetry of wetland depressions. The monotonic cubic spline and power
function are employed to fit the hypsometric relationships (i.e., depth-area relations) to derive the
above-water hypsometric curve $f_A(L)$ and the underwater hypsometric curve $f_B(L)$ (Messager et al.,
2016; Yao et al., 2018), respectively. Subsequently, the integration of these two curves (Figure 7) is
performed to calculate the WDWSC, represented as $V$ in Eq. 4:
$$V = \int_{h_{watersurface}}^{h_{max}} f_A(L)dL + \int_{0}^{h_{watersurface}} f_B(L)dL \qquad (4)$$

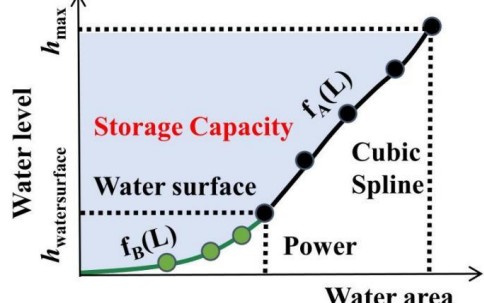


**Figure 7. Schematic diagram for the estimation of wetland depression water storage capacity. Two**
**depth-area rating curves are applied for the bathymetric volume and the above-water topographic volume.**

**3   Validation sites and datasets**
**3.1   Validation sites**

We applied the WetlandSCB to two wetlands in the Nenjiang River Basin (NRB), northeast China,

to validate the framework. Draining a total area of 297,100 km$^2$, the NRB is one of the largest river
basins in north China. In this river basin, agricultural lands and wetlands (lakes and swamps) are
prevalent (Wu et al., 2023). Recognised as critical regulators of the water balance within the NRB,
wetlands are considered more important than other ecosystems in mitigating future hydrological





extremes and increasing water availability for agriculture (Chen et al., 2020, Wu et al., 2020a, Wu et al.,
2020b, Wu et al., 2020c). For method validation and application of the WetlandSCB framework, we
focused on two national nature reserves within the NRB: the Baihe Lake and the Chagan Lake. The
Baihe Lake, characterised as a marsh wetland, covers approximately 40 km$^2$, predominantly
comprising seasonal inundation zones, with an average water depth of less than 1 m. In contrast, The
Chagan Lake is a large lacustrine wetland of about 372 km$^2$, mainly composed of perennial inundation
zones, with an average water depth of 2.5 m. These two validation wetlands represent different
characteristics in terms of type, area, and average water depth to verify the application robustness of
our developed framework. Field measurements of topographic and bathymetric information (elevation
and depth) were conducted for both the Baihe Lake and the Chagan Lake, consisting of 248 and 657
measurement points, respectively (Figure 8).

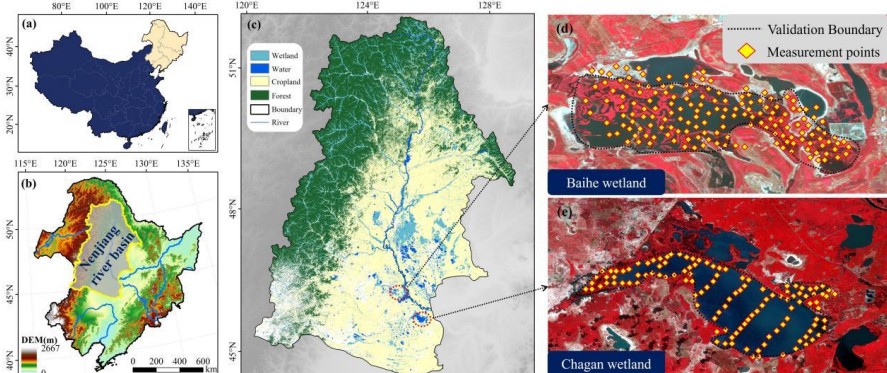


**Figure 8. Locations and distribution of elevation and depth measurements across the Baihe Lake and**
**Chagan Lake in the Nenjiang River basin, northeast China.**

**3.2   Datasets**

The application of the WetlandSCB framework requires the following data: (i) the global DEMs

sourced from SRTM DEM, with water distribution map sourced from the accompanying SRTM Water
Body Data (https://earthexplorer.usgs.gov/); (ii) wetland maps extracted from the 30-m resolution land
cover data for the years 1990-2019 (https://zenodo.org/records/5816591, Yang and Huang, 2021) and
30-m resolution wetland map in 2015 year (http://northeast.geodata. cn/index. html, Mao et al., 2020).
This study overlays the data from both sources to reduce the uncertainties in the wetland maps; (iii)
water distribution maps and water occurrence map obtained from the Global Surface Water datasets



(https://earthengine.google.com, Pekel et al., 2016); (iv) altimetry satellite data sourced from the
Sentinel-3A/3B products (https://scihub.copernicus.eu/). In addition, pre-processing of Sentinel-3
altimetry data is performed using the geophysical and atmospheric correction method developed by
Huang et al. (2019) (Eq. 5 and Eq. 6) to improve data accuracy:
$$H_{waterlevel} = H_{alt} - R - Cor \qquad (5)$$
where $H_{waterlevel}$ is the water level referenced to the EGM96 geoid, $H_{alt}$ is the altitude of the
altimeter derived from the modeling of satellite trajectory, $R$ is the range computed through the time
duration of the echoes, and $Cor$ is referred to as the geophysical and environmental corrections:
$$Cor = C_{dry} + C_{wet} + C_{iono} + C_{solidEarth} + C_{pole} + C_{EGM96} \qquad (6)$$
where $C_{dry}$, $C_{wet}$, $C_{iono}$, $C_{solidEarth}$, $C_{pole}$ and $C_{EGM96}$ are the dry tropospheric, wet tropospheric,
ionospheric corrections, the solid Earth tide, polar tide corrections and the EGM96 geoid respectively.
**4    Results and discussions**
**4.1    Performance evaluation of wetland depression spatial delineation and uncertainty analysis**
The performance of wetland depression spatial delineation based on the WetlandSCB framework
was evaluated using four indicators: overall accuracy, kappa coefficient, producer's accuracy, and user's
accuracy (Fig. 9a-f). The results indicate that the WetlandSCB framework can accurately determine the
spatial distribution of wetland depressions, with all four indicators exceeding 0.95. In contrast, the
user's accuracy is above 0.93 in both validation wetlands (error of commission is 0.07), and the
producer's accuracy is only 0.37 (error of omission is 0.63) in Baihe Lake based on the priority-flood
algorithm. The findings suggest that the algorithm can effectively identify wetland depressions, but is
limited by the numerical errors of the global DEMs, which leads to lower extraction accuracy of the
spatial distribution of wetland depressions (Zhou et al., 2016). In comparison, the WetlandSCB
framework outperforms the priority-flood algorithm in wetland depression spatial delineation.
Uncertainty in wetland depression spatial delineation using the WetlandSCB framework primarily
mainly arises from morphological operators and prior information on water distribution map. Figures
9g and 9h show that, compared with morphological operators, prior information on water distribution
map can significantly alter the performance of wetland depression spatial delineation and is a key
factor in determining the level of uncertainty. For instance, in Baihe Lake, the overall accuracy and
kappa coefficient improved by 0.29 and 0.56, respectively, after processing with prior information on

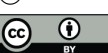

water distribution map. Similar studies have also found that the type and reliability of prior information
are major factors affecting the spatial filling performance of surface water maps (Aires, 2020;
Pulvirenti et al., 2011b). Therefore, this study compared the wetland depression spatial delineation
results based on three sets of prior information on water distribution map: GLC-FCS30 (from Zhang et
al., 2021), CLCD (from Yang and Huang, 2021), and JRC (Fig. 9i and 9j). The overall accuracy
differences for the Baihe Lake and Chagan Lake ranged from 0.68 to 0.98 and from 0.93 to 0.99,
respectively. In general, the accuracy levels of prior information from high to low were JRC >
GLC-FCS30 > CLCD. This suggests that selecting highly reliable prior information on water
distribution map is an essential way to reduce uncertainty in the WetlandSCB framework.

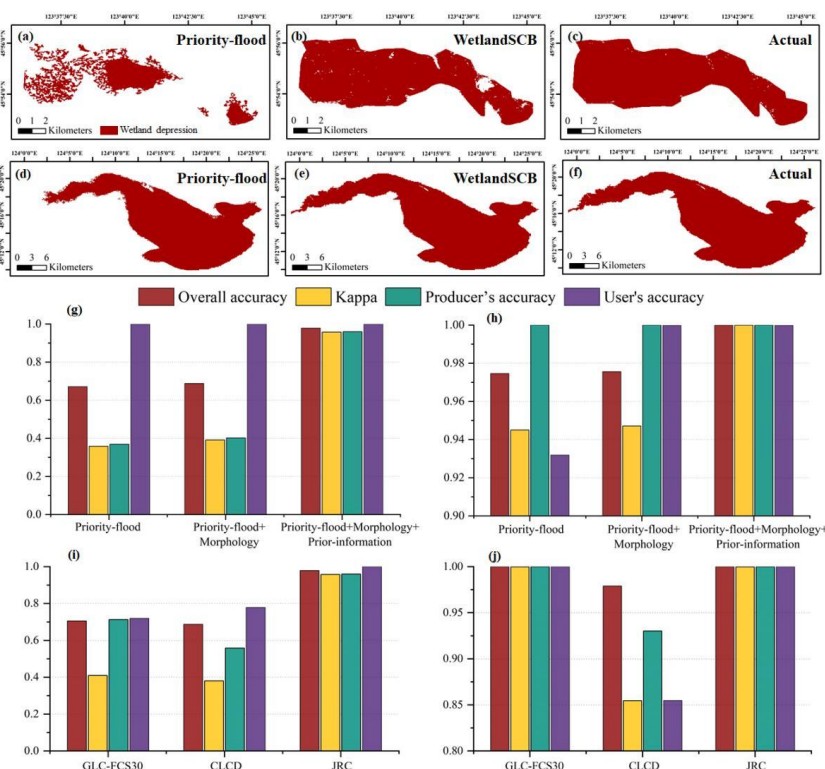


**Figure 9. (a), (b), and (c) depict the spatial distribution of wetland depressional areas in the Baihe Lake**
**based on the priority-flood algorithm, WetlandSCB framework, and field measurements, respectively. (d),**
**(e), and (f) show the corresponding results for the Chagan Lake. The impact of morphological operators and**
**prior information on water distribution map from the WetlandSCB framework is illustrated in (g) and (h).**
**The influence of different prior information on water distribution map from the WetlandSCB framework is**





presented in (i) and (j).

### 4.2   Performance evaluation of above-water topography correction and uncertainty analysis

The consistency between the original and corrected above-water topography and the actual
above-water topography obtained from field measurements can be evaluated using Pearson correlation
coefficients and $R^2$. The results indicate that the consistency between the original and actual
above-water topography is remarkably low, with $R^2$ values less than 0.2 for both validation wetlands.
Previous studies have also observed significant numerical discrepancies between the original and actual
above-water topography in some regions (e.g., Mukul et al., 2017; Uuemaa et al., 2020). Compared to
the original results, the consistency between the corrected and actual above-water topography
significantly improves. For example, the Pearson correlation coefficient and $R^2$ reach -0.74 and 0.55 in
the Baihe Lake, respectively, demonstrating that the WetlandSCB framework can effectively correct
numerical biases in above-water topography.

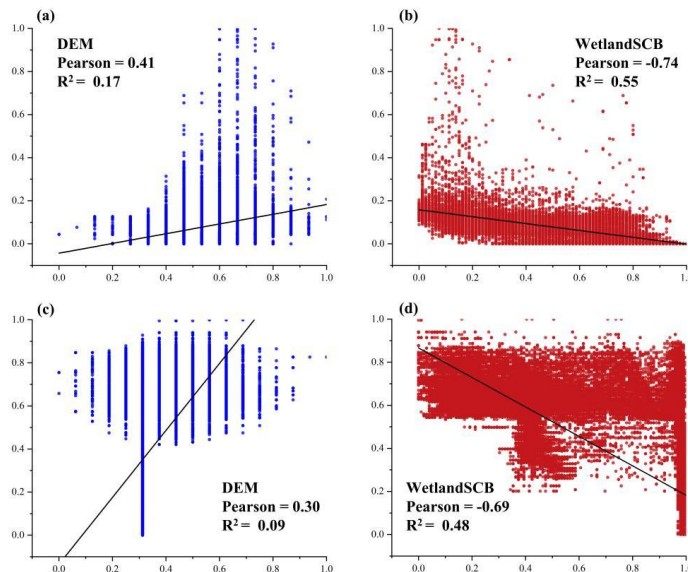


**Figure 10. (a) and (b) Consistency analysis results between the original and corrected above-water**

**topography for Baihe Lake. (c) and (d) are corresponding results for Chagan Lake.**


Uncertainty in correcting above-water topography using the WetlandSCB framework depends
primarily on the accuracy of the water occurrence map. Therefore, this study analyzed the correlation





between two sets of global-scale water occurrence maps, namely GLAD (Pickens et al., 2020) and JRC,
with actual above-water topography. The results show that the correlation level of GLAD is superior to
JRC in the Baihe Lake, while the opposite is observed in the Chagan Lake. Additionally, the $R^2$ values
for both sets of water occurrence maps are less than 0.4 (Figure 11c-f), which is significantly lower
than the accuracy level of the corrected above-water topography. This clearly shows the superiority of
the water occurrence map generated by the WetlandSCB framework over the GLAD or original JRC
map.

It is to note that the water occurrence map generated by the WetlandSCB framework still has a

certain level of uncertainty. First, the extraction of a complete and accurate water spatial distribution
from cloud-free images is constrained by factors such as the classification algorithm (Figure 11a)
(Peket et al., 2016), but some correction algorithms have been proposed to enhance raw water
distribution images (Zhao and Gao, 2018). Second, there is currently a lack of high-precision,
temporally and spatially continuous water distribution maps (Figure 11b). Future efforts could include
the use of image fusion methods, such as the Spatial and Temporal Adaptive Reflectance Fusion Mode,
to fuse data from multi-source remote sensing products such as Sentinel-2, MODIS, and Landsat,
which can effectively enhance the accuracy of water occurrence map (He et al., 2020; Wang et al.,

2016).

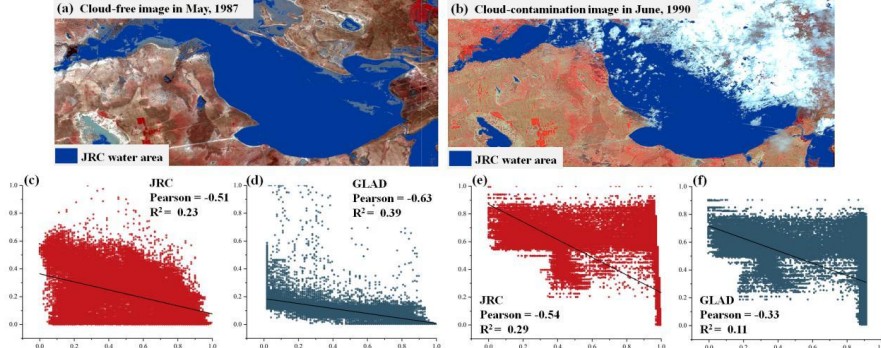


**Figure 11. (a) and (b) depict sources of uncertainty in water occurrence map generated by the WetlandSCB**
**framework. (c), (d), (e) and (f) illustrate the difference between two water occurrence maps on the**
**performance of above-water topography correction in the Baihe Lake and the Chagan Lake.**

**4.3    Performance evaluation of bathymetric information estimation**





356 The slope profile $p$ is used to describe the bathymetry of wetland depressional areas. The

357 calculated $p$ values for the Baihe lake and the Chagan Lake using the WetlandSCB framework are 7.45

358 and 4.08, respectively. The relative errors with respect to the actual $p$ values obtained from field

359 measurements are both less than 3%, demonstrating the high accuracy of the framework in estimating

360 underwater bathymetry.

361 To further prove the superiority of the WetlandSCB framework in estimating bathymetry, this

362 study employed spatial prediction and modeling methods, resulting in a p value of 8.65 for the Baihe

363 Lake and 4.78 for the Chagan Lake. The relative errors with respect to the actual $p$ values are both

364 greater than 18%, indicating that this method may lead to substantial errors in some regions, as also

365 reported by Papa et al. (2013) and Vanthof and Kelly. (2019). Furthermore, previous studies have often

366 applied smoothing methods to the global DEMs to enhance the accuracy of topographic

367 characterization in wetland depressions (e.g., Jones et al., 2018; Wu et al., 2019). In this regard, we

368 further used the Gaussian-smoothed global DEMs and the spatial prediction and modelling methods to

369 calculate $p$ for the Baihe Lake and the Chagan Lake. The resulting values were 8.51 and 4.37, with

370 relative errors of 17.63% and 7.9%, respectively. This underscores that smoothing methods do indeed

371 contribute to improving the accuracy of topographic information in wetland depressions. Notably, the

372 relative error for the Chagan Lake is significantly lower than that for the Baihe Lake, which is

373 consistent with the findings of Liu and Song (2022), who reported that the spatial prediction and

374 modeling methods are suitable for wetlands with long and narrow shape. In summary, it can be seen

375 that the WetlandSCB framework excels in the accuracy of estimating bathymetry in wetland

376 depressional areas when compared to other methods.

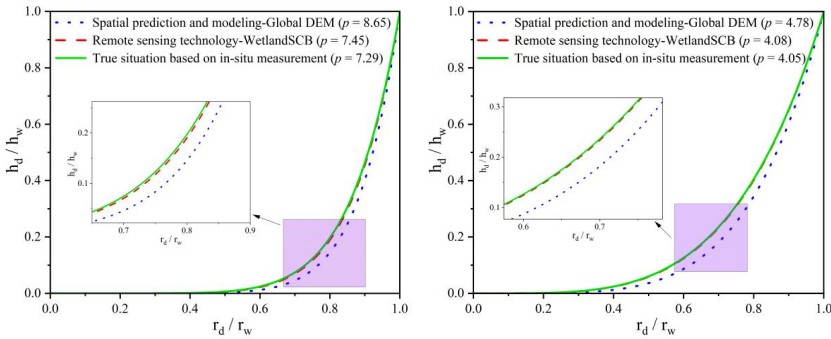


378 **Figure 12. Slope profile $p$ values of wetland depressions for the Baihe Lake (left) and the Chagan Lake**



(right), calculated with spatial prediction and modeling methods, and the WetlandSCB framework in
comparison with filed measurements.

**4.4 WetlandSCB framework application and implications for integrated water resources**
**management**
Wetland depressions are largely disregarded in many hydrologic modeling practices. Rare studies
exist on how their exclusion can lead to potentially inaccurate model projections and understanding of
hydrologic dynamics across the world's river basins (Rajib et al., 2020). This study applied a novel
framework delineating the topography and bathymetry of wetland depressional areas and focusing on
two distinctive wetlands to estimate WDWSC. Using the field measurements of topography and
bathymetry of the Baihe Lake and the Chagan Lake, the depth-area hypsometric curves were
constructed, and the WDWSC of the Baihe Lake and the Chagan Lake were estimated to be 61 million
m³ and 526 million m³, respectively (Fig. 13). The estimation results based on the WetlandSCB
framework were correspondingly 55 million m³ and 521 million m³, and the relative errors with the
actual measured WDWSC were both less than 10%, which is a good level of accuracy in estimation
precision (Moriasi et al., 2015). These results demonstrate the ability of the framework to accurately
estimate WDWSC, which can be applied to regions lacking field measurement data for global-scale
wetland water storage capacity estimation.

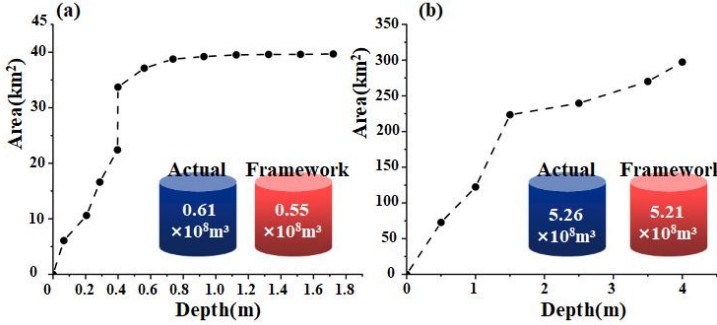


**Figure 13. The dashed line and blue cylinder represent the actual hypsometric curve and the corresponding**
**actual WDWSC based on field measurements, respectively. The red cylinder indicates the estimated**
**WDWSC from the WetlandSCD framework for the Baihe Lake (a) and the Chagan Lake (b).**



Wetlands play a pivotal role in mitigating flood and drought risks, as well as addressing water
scarcity challenge within a river basin. Previous studies underscore the significant impact of wetlands
in attenuating future flood characteristics, including peak flows, mean flows, duration, and flow
volume for various return period floods (Wu et al., 2023). Concurrently, wetlands contribute to
enhancing baseflow during both summer and winter seasons in the NRB (Wu et al., 2020c). Given the
NRB is a agriculture-dominated river basin, wetlands serves as the main water supply nodes by
collecting the flash flood and storing and purifying irrigation return flows. This reclaimed water can be
efficiently reused for irrigation purposes in the NRB (Meng et al., 2019; Smiley and Allred, 2011; Zou
et al., 2018). The WDWSC is a key parameter for evaluating the flood control and water supply
capacity of wetlands, also as a important prerequisite for understanding the impact of wetlands on
extreme hydrological events (Acreman and Holden, 2013). Therefore, the developed WetlandSCB
framework, which can provide accurate estimation of the WDWSC, contributes to the management of
food and water security in the NRB. Against the backdrop of global environmental change,
characterized by an escalation in the intensity and frequency of extreme hydrological events, and the
increasing disparity between water resource supply and demand, there is an urgent need for a novel
integrated water resources management approach based on natural solutions (Rodell and Li, 2023;
Thorslund et al., 2017; Yin et al., 2018). Wetlands have emerged as a nature-based solution in various
water resources management practices (Ferreira et al., 2023). Taking advantage of the wetland
hydrological regulation functions is instrumental in addressing the risks of flood and drought disasters
arising from global climate change, land use change, as well as the water scarcity risks stemming from
agricultural-ecological water competition. This can help develop effective adaptation strategies and
decisions for integrated water resources management.
**5   Conclusions**
This study developed a novel framework to accurately quantify wetland depression water storage
capacity using coarse-resolution terrain data. The developed framework, WetlandSCB integrates
multi-source remote sensing data, historical maps and prior knowledge, and achieved a high prediction
of wetland depressional distribution and water storage capacity. This is achieved through four steps: 1)
integrating priority-flood algorithm, morphological operators and prior information on water
distribution maps to delineate spatial extent of wetland depressional areas; 2) correcting numerical
biases in above-water topography with water occurrence map; 3) coupling spatial prediction and



modeling with remote sensing techniques to estimate bathymetric information, and 4) quantifying
depressional area water storage capacity based on depth-area rating curves. The concept and technical
approaches are applicable to large-scale wetland depression water storage estimation, as well as to the
regions where field measurements and/or high-resolution data are not available. Application of the
WetlandSCB framework provides accurate distribution and depth-area relations of wetland
depressional areas which can be incorporated into wetland modules of hydrological models (e.g.,
HYDROTEL, SWAT, HYPE, CHRM) to improve the accuracy of flow and storage predictions in river
basins.

*Data Availability.*

The data used in this study are openly available for research purposes. The SRTM DEM and

SRTM Water Body Data can be downloaded at https://earthexplorer.usgs.gov. Wetland maps are
available at https://zenodo.org/records/5816591 and http://northeast.geodata. cn/index. html. Water
distribution maps and water occurrence map are available at https://earthengine.google.com. Altimetry
satellite data can be downloaded at https://scihub.copernicus.eu.

*Author contribution.*
Boting Hu, Liwen chen and Yanfeng Wu designed and executed the study, all authors contributed to
general idea, the discussion and editing of the manuscript.

*Competing interest.*
The authors declare that they have no conflict of interest.

*Acknowledgments.*

This work was supported by the Strategic Priority Research Program of the Chinese Academy of

Sciences, China (XDA28020501), National Natural Science Foundation of China (41877160), National
Key Research and Development Program of China (2017YFC0406003, 2021YFC3200203) and The
Consulting Project Proposal of the Chinese Academy of Engineering (JL2023-17).

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
