# Peer review of "A novel framework for accurately quantifying wetland"

_Hydrology and Earth System Sciences, 2024_

## Referee Comment (RC2)

[referee-annotated manuscript omitted]

---

## Author Response (AR1)

**Reviewers' Comments and Authors Response**

Comments from Editor: Thank you for patience with the review of this manuscript. It took us a while to find two excellent reviewers. Both of the reviews are in. Based on the reviews it is clear that the manuscript much undergo revision. Please go ahead and submit your responses to the reviewers, I'll share the revised manuscript with the original reviewers for their recommendation. Thank you!

Authors' response: Many thanks for your dedication to the review process and giving us the opportunity to revise our manuscript. We appreciate it very much. I believe that the comments given by the reviewers have helped us to improve the quality of the manuscript. We have diligently revised the manuscript and provided a detailed point-by-point reply to the reviewer' comments. We would be grateful if you could consider my revised manuscript for the review process and hope that the current manuscript will meet the quality of Hydrology and Earth System Sciences.

Reviewer #1: This manuscript titled "A novel framework for accurately quantifying wetland depression water storage capacity with coarse-resolution terrain data" by Hu et al. presents a new framework towards better quantification of wetland depression water storage capacity, a critical component contributing to wetland hydrology. This proposed framework is capable to infer wetland water storage capacity with high accuracy and its statistical distribution from multisource information, including satellite data, historical maps, and prior knowledge. From my perspective, the research issue that this manuscript is targeting is interesting and requires urgent attention; moreover, the proposed framework can bring expertise to address this research question. Whereas I found a little difficult to follow in certain parts, where I think the authors need to further revise before the manuscript is ready to publish.

Authors' response: Thank you very much for taking the time to review our manuscript and providing valuable and constructive comments. We greatly appreciate your insightful comments, which have been instrumental in helping us improve the quality of the manuscript. In response, we have diligently revised the manuscript (highlighted in yellow) according to your comments and suggestions.

1. Firstly, I feel some disconnection between identified research questions and the proposed method. In line 207, the authors mention removing outliers from underwater area-level pairs. The concept of underwater area-level pairs seems to come to a sudden without much preparation. I' m confused about how do the authors obtain the underwater area-level pairs, from existing approaches like spatial prediction and remote sensing as reviewed in lines 88-117, or from some author-developed methods? If they come from existing approaches, wouldn' t it be a conflict since the authors just identified critical shortcomings to address in the review paragraph (lines 88-117)? Or if these area-level pairs are obtained from some approach developed by the authors, the method should be mentioned and details should be provided, either in the manuscript or supplemental material.

Authors' response: Thank you for this valuable comment. We fully agree that the explanation of the underwater bathymetric estimation method was insufficient. Below, we provide our responses and revisions.

The absence of underwater bathymetric information in global DEMs is primarily due to the water distribution at the time of data acquisition. Taking SRTM DEM data as an example, the water extents in 2000 define the spatial extent of the water mask, which excludes land surface elevation measurements in the SRTM DEM data. With advancements in satellite radar/laser altimetry, data from Envisat, ICESat, CryoSat, Jason-1/2/3, SARAL, and Sentinel-3 have been available from 1992 to the present. Consequently, for each wetland, numerous water levels below the 2000 reference water level have been recorded. This enables the use of remote sensing technologies to construct underwater area-level pairs.

However, altimetry satellite data are subject to various factors that influence the accuracy of water level monitoring. These include intrinsic factors such as sensor performance and instrument resolution, as well as extrinsic factors like natural elements (e.g., clouds and wind), the geometry of the wetland water body, boundary conditions, and vegetation characteristics (Donlon et al., 2012; Gao et al., 2019; Zhou et al., 2023). Consequently, the derived water level data exhibit substantial variability and uncertainty. For instance, Figure 1 presents the water level variations extracted from two altimetry satellites, Sentinel-3 and ICESat-2, for the Huaao wetland in the Nenjiang River Basin, where water level differences

during the same period can reach up to 2 meters. Relying solely on mathematical methods (e.g., the coefficient of variation) or on natural rules (e.g., water levels in wet seasons being higher than those in dry seasons) proposed in previous studies is insufficient for accurately determining water level values.

To address the challenge, this study proposes an improved method for estimating underwater topography in wetland depressions by integrating spatial prediction and remote sensing techniques. The method is based on the assumption that the relationship between water surface area and water level is unique and that the slope of the above-water topography is generally continuous with that of the underwater topography. By systematically iterating through combinations of water surface area and water level, we identify the combination that best matches the slope of the above-water topography, as corrected by the WetlandSCB framework, as the optimal solution for characterizing the underwater topography of wetland depressions.

Figure 1. (a) Huaao wetland depression location. (b) water level data extraction from Sentinel-3 and ICESat-2 altimetry satellites.

To address the discontinuity of the manuscript, I have added the following content before line 207: "Match multi-source altimetry satellites with optical images to construct all area-level pairs for wetland depressions. By identifying water surface distributions in global DEMs, filter the area-level pairs that represent underwater hypsometric relationships within wetland depressions."

**References:**

Donlon, C., Berruti, B., Buongiorno, A., Ferreira, M.H., F' em' enias, P., Frerick, J., Goryl, P.,

and Sciarra, R.: The global monitoring for environment and security (GMES) sentinel-3 mission. Remote Sens. Environ. 120, 37-57. https://doi.org/10.1016/j.rse.2011.07.024, 2012. Gao, Q., Makhoul, E., Escorihuela, M.J., Zribi, M., Quintana Seguí, P., García, P., and Roca, M.: Analysis of retrackers' performances and water level retrieval over the Ebro River basin using sentinel-3. Remote Sens. 11 (6), 718. https://doi.org/10.3390/rs11060718, 2019. Zhou, H., Liu, S., Mo, X., Hu, S., Zhang, L., Ma, J., Bandini, F., Grosen, H., and BauerGottwein, P.: Calibrating a hydrodynamic model using water surface elevation determined from ICESat-2 derived cross-section and Sentinel-2 retrieved sub-pixel river width. Remote Sens. Environ. 298, 113796. https://doi.org/10.1016/j. rse.2023.113796, 2023.

2. Another major comment is related to the discussion of the application of this proposed framework for integrated water resources management (Section 4.4). I agree that the wetland depressions are largely disregarded in hydrological models and that this proposed framework could bring possible improvements towards hydrological and water resources simulations, but by what means? The entire section is focused on the application and implication, but other than describing how it is important to account for wetland depression and the potential improvement it can bring, the authors do not depict a picture on how to fit the proposed framework into hydrological modeling and water resources management. To make the proposed framework more accessible to its potential clients (or to better sell the framework to hydrological modeling community), I suggest adding languages on how-to for the application part. If the authors have yet had a specific way for the application, it would also help by just providing general procedures. Additionally, a flowchart showing the application way will also be more than helpful.

Authors' response: Thank you for this valuable suggestion. Over the past three months, I have given it considerable thought and taken actions to address it. The VIC-5 model incorporates a specialized lake/wetland module, and I have precisely prepared input parameters based on the WetlandSCB framework, such as wetland hypsometric relationships, wetland catchment area, and other module-relevant parameters. Furthermore, I have refined the parameters of wetland soil and vegetation parameters (e.g., wetland soil depth is approximately 1 meter, and the dominant vegetation is reeds in the Nenjiang River Basin). Using these inputs, I conducted simulations to evaluate the impacts of wetland dynamics on

soil moisture and runoff. The results exhibited a high degree of accuracy, highlighting the strong potential of the WetlandSCB framework to enhance wetland eco-hydrological modeling studies.

Figure 2. Accuracy assessment of the VIC hydrological model with lake/wetland module in the Nenjiang river basin (5km spatial resolution).

To provide guidance on "how-to for the application", I have added the following content and a flowchart in Section 4.4: "Using the WetlandSCB framework, raster-scale wetland depression topographic information can be accurately reconstructed. Through flow direction analysis and watershed delineation methods, key parameters such as wetland inflow and outflow locations, wetland catchment boundaries, and other related characteristics can be identified (these steps can be performed using QGIS software). By integrating the hypsometric curve, water surface distribution data, and morphological characteristics of the wetland derived from the WetlandSCB framework, the initial wetland water level, the number of wetland layers, and the corresponding area–level pairs can be determined. Field surveys provide essential data on wetland soil and vegetation properties as well as inflow volumes within the study area. Finally, the hydrological model, coupled with the wetland module, can be implemented to support wetland eco-hydrological research and integrated water resources management."

Figure 3. Integration process and application outputs of the WetlandSCB framework with VIC hydrological model.

**In terms of minor comments:**

1. [Line 75-76] The reviewer thinks it necessary to provide citations to support this claim.

Authors' response: Thank you for this comment. We have provided the following references.

Chen, T., Song, C., Zhan, P., Yao, J., Li, Y., and Zhu, J.: Remote sensing estimation of the flood storage capacity of basin-scale lakes and reservoirs at high spatial and temporal resolutions, Sci. Total Environ, 807, 150772, https://doi.org/10.1016/j.scitotenv.2021.150772, 2022.

Liu, K., Song, C., Zhao, S., Wang, J., Chen, T., Zhan, P., Fan, C., and Zhu, J.: Mapping inundated bathymetry for estimating lake water storage changesfrom SRTM DEM: A global investigation. Remote Sens. Environ, 301, 113960, https://doi.org/10.1016/j.rse.2023.113960, 2024.

Liu, K., Song, C., Ke, L., Jiang, L., Pan, Y., and Ma, R.: Global open-access DEM performances in Earth's most rugged region High Mountain Asia: Amulti-level assessment. Geomorphology, 338, 16-26, https://doi.org/10.1016/j.geomorph.2019.04.012, 2019.

2. [Line 207] A typo in "are-level pairs".

Authors' response: Thank you for this comment. We have revised it to "area-level pairs".

Reviewer #2: The authors propose a remote sensing-based method to determine wetland depression water storage capacity that includes four steps: 1) spatial delineation of wetlands, 2) reconstruction of above-ground topography, 3) bathymetric estimation, and 4) storage capacity estimation. For validation, the method was applied to two lakes in the Nanjiang River basin in China with transects of field-measured topography and bathymetry. Wetlands play an important role in modulating streamflow and understanding their storage characteristics is critical to predicting both water supply and flood events. The authors address an important question: how to assess wetland depression water storage capacity in areas without high resolution digital elevation models (DEMs).

Authors' response: First of all, we want to thank the reviewer for taking time to review our manuscript. We appreciate the reviewer's constructive comments and suggestions, all of which have been taken into account in the revised manuscript (highlighted in green) as described below.

However, the manuscript has several major deficiencies, including:

1) The methods section requires substantial revision as it does not describe the proposed method to estimate wetland depression storage capacity with sufficient clarity and detail that it could be reproduced independently. It should describe the methods used in the research described in the manuscript, not past studies (e.g., the figures show examples from other papers). Section 3.2 lists several datasets used in the study, but the methods section does not describe how they are used. Most importantly, the manuscript does not explain how the authors actually calculated a volumetric estimate of wetlands depression water storage capacity.

Authors' response: As the reviewer suggested, we added more details description on the approach that we used to estimate depression storage. First, we elaborated on the operational details of various methods and algorithms, with a particular emphasis on the specific datasets employed. Additionally, we supplemented schematic diagrams to illustrate the methodological framework and wetland characteristics within the Nenjiang River Basin. Finally, we described the application of hypsometric curves to clarify the computational approach for determining wetland depression water storage capacity.

2) The results and discussion section also requires substantial revision. The section does not provide enough detail on the method performance assessment to evaluate the method's skill.

Authors' response: As the reviewer suggested, we provided a more detailed description to evaluate the performance of the framework. First, we supplemented the introduction of evaluation metrics. Then, we refined the description of the WetlandSCB framework's superior accuracy in delineating wetland depressions compared to the priority-flood algorithm, as well as its higher precision in estimating bathymetric curves compared to spatial prediction and modeling methods. Additionally, we incorporated a comparative analysis of wetland depression water storage capacity estimates derived from the WetlandSCB framework and other similar research approaches.

3) The conclusions section should state the metrics used to quantify the method's performance.

**Authors' response:** As suggested, we modified the Conclusions by adding the performance metrics to demonstrate the applicability of the method. We provided a detailed description of the accuracy metrics of the WetlandSCB framework in wetland depression delineation, above-water topography correction, bathymetric estimation, and wetland depression water storage capacity calculation, highlighting its superiority over other methods.

Specific comments are included in the supporting document.

Line 46 - This is a sentence fragment

Authors' response: Thank you for this comment. The sentence has been modified as "The quantitative study of the WDWSC contributes to advancing scientific understanding of wetland hydrological regulation functions and to improving integrated water resources management at the watershed scale."

Line 49 - Please include a technical description of the wetland depression identification algorithms.

*Authors' response*: Thank you for this comment. As the following sentence has been added: "In a vector-based contour representation, wetland depressions are shown as nested closed contours, with inner contours at lower elevations than the outer ones (Wu and Lane, 2016)."

Line 51 - Please include a more detailed description of this step.

Authors' response: Thank you for this comment. The sentence has been modified as "Area-depth pairs are derived from the contour lines of wetland depressions, and hypsometric curves are constructed by applying curve-fitting methods to the obtained pairs."

Line 57 - This statement needs citations.

Authors' response: Thank you for this comment. Added the following reference: "Wu, Q., Lane, C. R., Wang, L., Vanderhoof, M. K., Christensen, J. R., and Liu, H.: Efficient delineation of nested depression hierarchy in digital elevation models for hydrological analysis using level-set method, J. Am. Water Resour. Assoc, 55(2), 354-368, https://doi.org/10.1111/1752-1688.12689, 2019."

Line 77 - The text above describes inaccuracies in the above-water DEMs. Please include an explanation of the bias (defined in statistics as a systematic tendency).

Authors' response: Thank you for the suggestion. We modified the sentence and added the following two references: "The above-water DEMs demonstrate systematic overestimation caused by canopy height, and their accuracy is significantly influenced by terrain slope (Marešová et al., 2024; Simard et al., 2024)."

**References:**

Marešová, J., Bašta, P., Gdulová, K., Barták, V., Kozhoridze, G., Šmída, J., Markonis, Y., Rocchini, D., Prošek, J., Pracná, P., and Moudrý, V.: Choosing the optimal global digital elevation model for stream network delineation: Beyond vertical accuracy, Earth Space Sci, 11(12), p.e2024EA003743, https://doi.org/10.1029/2024EA003743, 2024.

Simard, M., Denbina, M., Marshak, C., and Neumann, M.: A global evaluation of radar-derived digital elevation models: SRTM, NASADEM, and GLO-30, J. Geophys. Res.-Biogeosci, 129(11), e2023JG007672, https://doi.org/10.1029/2023JG007672, 2024.

Line 79 - Citations are needed for these statements.

Authors' response: Thank you for the suggestion. Added the following references:

Gdulová, K., Marešová, J., and Moudrý, V.: Accuracy assessment of the global TanDEM-X

digital elevation model in a mountain environment, Remote Sens. Environ, 241, 111724, https://doi.org/10.1016/j.rse.2020.111724, 2020.

Hawker, L., Neal, J., and Bates, P.: Accuracy assessment of the TanDEM-X 90 digital elevation model for selected floodplain sites, Remote Sens. Environ, 232, 111319, https://doi.org/10.1016/j.rse.2019.111319, 2019.

Li, S., MacMillan, R. A., Lobb, D. A., McConkey, B. G., Moulin, A., and Fraser, W. R.: Lidar DEM error analyses and topographic depression identification in a hummocky landscape in the prairie region of Canada, Geomorphology, 129(3-4), 263-275, https://doi.org/10.1016/j.geomorph.2011.02.020, 2011.

Liu, K., Song, C., Zhao, S., Wang, J., Chen, T., Zhan, P., Fan, C., and Zhu, J.: Mapping inundated bathymetry for estimating lake water storage changesfrom SRTM DEM: A global investigation, Remote Sens. Environ, 301, 113960, https://doi.org/10.1016/j.rse.2023.113960, 2024.

Figure 1d has a good description. Figures 1a-1c need more explanation. Where are the discrepancies in above-water topography? What is the bias? Where is the incompleteness and inaccuracy of wetland depression

Authors' response: Thank you for the suggestion. To provide the information, we redraw Figure 1 with a unified legend to represent the numerical differences between the LiDAR DEM and global DEMs. The spatial distribution of wetland depressions extracted using the priority-flood algorithm and global DEMs is compared with satellite imagery to characterize the incompleteness and inaccuracy of the wetland depressions.

Figure 1. Wetland depression extraction based on the priority-flood algorithm and global DEMs suffers from the bias of above-water topography(Figures 1a-1c illustrate the discrepancies in above-water topography between LiDAR DEM and ALOS DEM, where Figure 1a shows the 1m spatial resolution LiDAR DEM, Figure 1b displays the LiDAR DEM resampled to 30m spatial resolution using the nearest-neighbor method, and Figure 1c presents the 30m spatial resolution ALOS DEM), incompleteness and inaccuracy of wetland depressions identification(Figure 1d shows a historical satellite image from 2013, and Figure 1e depicts the spatial distribution of wetland depressions extracted using the priority-flood algorithm and ALOS DEM, which exhibits noticeable characteristics of incomplete boundaries and spatial fragmentation), and the absence of bathymetric information (Figure 1f, where the entire water surface is represented by a single elevation value of 129 m).

**Line 100 - citation?**

Authors' response: Two references were added:

Khazaei, B., Read, L. K., Casali, M., Sampson, K. M., and Yates, D. N.: GLOBathy, the global lakes bathymetry dataset, Sci. Data, 9(1), 36, https://doi.org/10.1038/s41597-022-01132-9, 2022.

Zhan, P., Song, C., Luo, S., Liu, K., Ke, L., and Chen, T.: Lake level reconstructed from DEM-based virtual station: Comparison of multisource DEMs with laser altimetryand UAV-LiDAR measurements, IEEE Trans. Geosci. Remote. Sens, 19, 1-5, https://doi.org/10.1109/LGRS.2021.3086582, 2021.

Line 106 - ICESat-2 should be included.

*Authors' response*: As suggested, we revised the sentence as "(e.g., Sentinel-3, CryoSat-2, ICESat-2, Envisat)"

Line 113 - The incompleteness and inaccuracies referenced here should be described in the previous text and The previous paragraphs indicate that the above-water topography is accurately represented in LIDAR DEMs. This statement contradicts that.

Authors' response: Thank you for this comment. We moved the content from lines 110-117 to the next paragraph; deleted the sentence "previous studies have mainly utilized LiDAR DEM data to estimate WDWSC," and modified the sentence to "In summary, studies using the global DEMs have overlooked critical issues such as the incompleteness and inaccuracy of wetland depression identification, as well as biases in above-water topography, leading to significant uncertainties in WDWSC estimation."

Line 138 - The figures in the methods should demonstrate the successfully execution of the steps on the site area, not other regions.

Authors' response: Thank you for the suggestion. Figures 3b and 3C have been replaced with a schematic diagram of representative wetland depressional areas located in the Nenjiang River Basin. For Figure 5, we added an elevation map, a water occurrence map of a wetland depressional area within the Nenjiang River Basin, and a correlation diagram to illustrate the relationship between the two maps.

Figure 3. (b) representative wetland depressional area located in Nenjiang river basin, China. (c) 3-dimensions diagram of wetland depressional areas.

Figure 5. (d) SRTM DEM and (e) JRC water occurrence map of a representative wetland depressional area located in the Nenjiang River Basin. (f) Correlation between elevations and water occurrences in the wetland depressional area.

Line 139 - The text should state the DEMs used.

Authors' response: "global DEMs" has been replaced with "SRTM DEM".

Line 140 - The text should specify the algorithm used.

Authors' response: As suggested, we added the sentence as "The priority-flood algorithm was applied to identify and fill sinks in the DEM, resulting in a depressionless DEM. By subtracting the original DEM from the depressionless DEM, an elevation difference grid was generated, where each cell value represents the depth of the depression. Subsequently, cells with elevation changes greater than zero were extracted and identified as topographic

depressions."

Line 149 - How were these indicators used to identify and exclude artifact wetland depressions?

Authors' response: Due to the distinct morphological differences between artifact wetland depressions and true wetland depressions, artifact depressions can be effectively removed by determining the optimal threshold for morphological indicators. We added the following text to clarify: "Artifact wetland depressions (e.g., rivers and channels) typically exhibit low circularity and high eccentricity, whereas true wetland depressions generally display high circularity and low eccentricity. By iteratively refining the threshold values of these indicators and validating the results through visual inspection, the optimal thresholds were established to effectively eliminate artifact wetland depressions."

Line 155 - There should be a more detailed technical description of how these operators are applied.

Authors' response: Thank you for this comment. We added the following statement as "On the Python platform, morphological opening was performed by first applying the erosion operator, followed by the dilation operator. These operations require a binary-valued kernel, where the output pixel value in the erosion step is determined by the minimum value within the kernel. A disk-shaped kernel with a 3-pixel radius was used, which is significantly smaller than typical wetland depressions but sufficient to eliminate speckle noise."

Line 160 - What is the water distribution map? This hasn't been defined.

Authors' response: The water distribution map is intended to show the maximum water body over the study region. We add the following sentence to clarify: "The open water distribution map is defined as the maximum water body distribution map."

Line 162 - What do you mean by "intersected"?

Authors' response: The "intersected" refers to the spatial union of the wetland depression map and the water distribution map. We modified the following sentence to clarify: "After applying morphological operators, the wetland depression map is merged with the water

distribution map within the depression boundaries through a union operation, ensuring the creation of a comprehensive and finalized wetland depression map."

Line 167 - This figure and caption needs substantial revision to make clear what it represents - e.g., what is "two depressions extent?", what do the colors represent?, what is the red dotted polygon?

*Authors' response:* Thank you for the suggestion. We revised both the figure and its caption entirely.

Figure 4. The final wetland depression map derived from morphological operators and prior water distribution information. Figure 4a depicts the spatial distribution of the wetland depression before processing, with pink indicating wetland depression pixels; Figure 4b shows the spatial distribution of the wetland depression after morphological operator processing, represented in white; Figure 4c illustrates the maximum water extent within the wetland depression boundaries, highlighted in blue; and Figure 4d presents the refined spatial distribution of the wetland depression, obtained by combining Figures 4b and 4c through a union operation. The red dotted polygons indicate wetland depression pixels supplemented with prior information.

Line 170 - A water occurrence map does not show elevation information and therefor cannot describe topography.

**Authors' response**: Thank you for this comment. We removed the statement ("The water occurrence map can effectively describe three-dimensional topography at a large spatial scale") and replace it with "The basic idea is that the greater the water occurrence for a pixel (i.e., the more frequently it is covered by water), the deeper the water (Li et al., 2021)."

Line 178 - This is a sentence fragment.

Authors' response: Thank you for this comment. The sentence has been rewritten in the revised manuscript: "However, the cloud-free JRC water distribution images have temporal discontinuity. They are more available during dry seasons than wet seasons, which leads to deviations in the representation of real topography at the scale of individual wetland depression."

Line 184 - reference? There are several cloud screening algorithms at GEE.

Authors' response: Thank you for this comment. We added a reference in the sentence: "using the cloud screening algorithm(A rudimentary cloud-scoring algorithm called simpleCloudScore (Mullen et al., 2021)."

Mullen, C., Penny, G., and Müller, M. F.: A simple cloud-filling approach for remote sensing water cover assessments, Hydrol. Earth Syst. Sci., 25, 2373-2386, https://doi.org/10.5194/hess-25-2373-2021, 2021.

Line 185 - reference?

*Authors' response*: Added the following reference: "Canny, J: A computational approach to edge detection, IEEE Trans. Pattern Anal. Mach. Intell., (6), 679-698, https://doi.org/10.1109/TPAMI.1986.4767851, 1986."

Line 190 - This will only work with large water bodies that are partially occluded by clouds (e.g., the Lake Mead example). It's not likely to be useful for small wetlands.

Authors' response: Thank you for your careful reading. Since the Canny algorithm requires a minimum of 9 pixels to delineate edges a closed curve necessitates at least 4 pixels, inferring that the smallest feasible wetland depressional area should encompass at least 16 pixels. Given that commonly used medium-resolution remote sensing data have a spatial resolution of 30m, this corresponds to an area of 0.0144 km2. Therefore, we added the following statement to explain: "Theoretically, this method can be applicable to wetland depressional areas exceeding 0.0144 km2."

Line 192 - What is the source of the elevation data?

*Authors' response*: The elevation data refers to medium-resolution global DEMs. Since this study specifically uses the SRTM DEM for validation, we rephrased the sentence to: "This is because the water occurrence values within the same wetland depression correspond to the elevation values of SRTM DEM."

Line 198 - Figure 5a appears to show areal imagery, not a JRC could contaminated JRC images and Missing description of 5c.

Authors' response: Thank you for catching this up. We revised the sentence to "Restoration method of cloud-contaminated satellite images." and add the sentence as "LiDAR DEM (b) and JRC water occurrence map (c) of Mead Lakes in the United States."

Line 202 - This section needs substantial revision. It should clearly and concisely describe the datasets and methods used by the authors to derive bathymetry. This is cannot be understood from the manuscript text.

Authors' response: Thank you for this comment. We revised the first paragraph as "Using remote sensing technologies, simultaneous observations of water areas provided by optical images (e.g., Global Surface Water datasets) and the corresponding water levels from altimetry satellites (e.g., Sentinel-3) are employed to obtain underwater area-level pairs. Furthermore, based on the principle of spatial prediction and modeling methods, the continuity of the slope profile between the above-water and underwater topography is used as a filtering criterion to refine the underwater area-level pairs, enabling precise characterization of the underwater topography of wetland depressions."

Line 207 - This should be defined.

Authors' response: Thank you for this comment. We added the following statement to clarify "Since altimetry satellite data are subject to various factors that influence the accuracy of water level monitoring, including intrinsic factors such as sensor performance and instrument resolution, as well as extrinsic factors like natural elements, the geometry of the wetland water body, boundary conditions, and vegetation characteristics (Zhou et al., 2023),

some of the derived water level data exhibit substantial variability and uncertainty and are regarded as outliers."

Zhou, H., Liu, S., Mo, X., Hu, S., Zhang, L., Ma, J., Bandini, F., Grosen, H., and BauerGottwein, P.: Calibrating a hydrodynamic model using water surface elevation determined from ICESat-2 derived cross-section and Sentinel-2 retrieved sub-pixel river width. Remote Sens. Environ. 298, 113796. https://doi.org/10.1016/j. rse.2023.113796, 2023.

Line 228 - This section describes how the authors derived a mathematical relationship between water level and water area. It does not describe how the authors estimated wetland depression storage capacity.

**Authors' response**: Thank you for this comment. We added the statement to explain the calculation process of wetland depression storage capacity: "Based on the underwater hypsometric curve  $f_B(L)$ , the area enclosed by the water level from 0 to the maximum value and  $f_B(L)$  is defined as the underwater storage capacity of the wetland depression. Similarly, based on the above-water hypsometric curve  $f_A(L)$ , the area enclosed by the water level from the minimum value (corresponding to the maximum value of  $f_B(L)$ ) to the maximum value (the elevation of the spilling point) and  $f_A(L)$  is defined as the above-water storage capacity of the wetland depression. The total wetland depression storage capacity is then obtained as the sum of both components."

Line 229 - This is a sentence fragment.

Authors' response: Thank you for this comment. We modified the sentence to "We derived the hypsometric relationship from the corrected above-water area-level pairs and estimated underwater area-level pairs of wetland depressions."

Line 256 - The text should specify the methods used to collect the field data.

Authors' response: Thank you for the suggestion. We added the statement and figure to introduce the filed data: "we combined an ultrasonic echo sounder (D390, Chcnav, China) with a Global Positioning System (GPS) positioning system and applied the field measurements according to the sectional method. Manned vessels in areas of greater water

depth and unmanned remotely operated vessels in areas of lower water depth with the aid of water rulers and hammers."

unmanned remotely operated vessels.

Line 264 - Citations should be included for the DEM and water distribution map.

Authors' response: Thank you for this comment. We added the following reference: "Farr, T. G., and M. Kobrick.: Shuttle Radar Topography Mission produces a wealth of data, Eos Trans. AGU, 81:583-583, https://doi.org/10.1029/EO081i048p00583, 2000." and "NASA JPL.: NASA Shuttle Radar Topography Mission Water Body Data Shapefiles & Raster Files [Data set]. NASA EOSDIS Land Processes Distributed Active Archive Center, Accessed 2025-02-04, https://doi.org/10.5067/MEaSUREs/SRTM/SRTMSWBD.003, 2013."

Line 283 - The text should specify what datasets are being compared to assess the accuracy of the wetland depressions spatial delineation.

Authors' response: Thank you for the suggestion. We added the following statement to clarify the compared datasets: "The actual topographic and bathymetric information obtained from field measurements, along with the contour-tree method, provides the actual spatial distribution of wetland depressional areas. Additionally, two spatial distributions of wetland depressional areas are derived: one using the SRTM DEM combined with the priority-flood algorithm and the other using the SRTM DEM with the WetlandSCB framework. A comparative analysis of these three approaches is conducted to assess the accuracy differences in wetland depression spatial delineation."

Line 290 - The text should specify how this was assessed.

Authors' response: Thank you for this suggestion. We added the statement as "Since the overall accuracy of wetland depression spatial delineation derived using the priority-flood algorithm exceeds 0.6 for both validation wetland sites, with a peak accuracy of 0.97 for Chagan Lake, the results demonstrate that the algorithm is highly effective in identifying wetland depressions."

Line 294 - This comparison should be included in the main text or in supplemental material.

Authors' response: Thank you for this comment. We added the statement as "Since the overall accuracy, Kappa coefficient, and Producer's accuracy of wetland depression spatial delineation obtained using the WetlandSCB framework show significant improvements over those derived from the priority-flood algorithm for both validation wetlands, with a slight increase in User's accuracy for Chagan Lake, the results effectively demonstrate that the WetlandSCB framework outperforms the priority-flood algorithm in wetland depression spatial delineation."

Line 303 - The GLC-FCS30 and CLCD datasets should be described in the methods section.

*Authors' response:* Thank you for this comment. We provided an introduction to the two datasets: "The water distribution map is derived from GLC-FCS30, CLCD or JRC data, where GLC-FCS30 and CLCD are 30-meter resolution land cover datasets, and JRC provides 30-meter resolution water surface data." in the methods section.

Line 304 - The text should describe how these metrics were calculated.

Authors' response: Thank you for the suggestion. We added the statement to describe the metrics: "The confusion matrix, also known as an error matrix, is a crucial method for evaluating land cover classification accuracy. It intuitively reflects the classification relationship between the evaluated data and the reference data. Key evaluation metrics include the Kappa coefficient, overall accuracy, producer's accuracy, and user's accuracy, among others. For detailed calculation equations, refer to Liu et al. (2007)."

Liu, C., Frazier, P., and Kumar, L.: Comparative assessment of the measures of thematic

classification accuracy, Remote Sens. Environ, 107(4), 606-616, https://doi.org/10.1016/j.rse.2006.10.010, 2007.

Line 309 - The text should specify the source of the "actual" wetland depression map.

Authors' response: Thank you for this comment. We added the following sentence in caption of Figure 9: "The actual wetland depression map was derived from field measurements using the contour-tree method."

Line 319 - The text should define the dataset the authors use as the "original" results.

**Authors' response**: Thank you for this comment. We added the sentence to define the original results: "Consider the elevation information directly obtained from the SRTM DEM as the original above-water topography."

Line 324 - The Pearson correlation is worse in the corrected topography than in the original, contradicting this statement. Negative correlation indicates both variables move in the opposite direction. This indicates very poor correlation between the two sources of topographic information.

Authors' response: We apologize for the incorrect expression in this figure. Due to the negative correlation between water occurrences and elevations, the data in the original figure was reversed. Figure 10 has been corrected, and the negative Pearson correlation coefficient in the original text has been changed to a positive value.

Figure 10. (a) and (b) Correlation analysis results between the original and corrected above-water topography for Baihe Lake. (c) and (d) are corresponding results for Chagan Lake.

Line 328 - The figure units should be defined.

**Authors' response**: Thank you for this comment. We added the statement in the caption "The elevation values are mapped to [0, 1] based on extreme value normalization."

Line 333 - The GLAD dataset should be described in the methods section.

Authors' response: Thank you for this comment. We added the statement in the methods section "The Global Surface Water Dynamics, produced by the Global Land Analysis & Discovery (GLAD), includes a water occurrence map (Pickens et al., 2020)."

Line 334 - Water occurrence maps do not contain topographical information, so it is not possible to compare them to topography. The negative Pearson's correlation indicates very poor correlation for each dataset.

Authors' response: Thank you for this constructive comment. Given that higher water occurrence values indicate deeper water, which typically corresponds to lower elevations (Armon et al., 2020; Li et al., 2021), our primary objective is to demonstrate that the corrected water occurrence map exhibits a higher correlation with the actual elevations. This characteristic is also validated in Figure 5f. Accordingly, we revised the statement to: "Due to the negative relationship between water occurrence values and elevations in wetland depressions, this study compared the correlation differences between two sets of global-scale water occurrence maps, namely GLAD and JRC, and the actual above-water topography of two wetland depressions."

Armon, M., Dente, E., Shmilovitz, Y., Mushkin, A., Cohen, T. J., Morin, E., and Enzel, Y.: Determining bathymetry of shallow and ephemeral desert lakes using satellite imagery and altimetry, Geophys. Res. Lett, 47(7), e2020GL087367, https://doi.org/10.1029/2020GL087367, 2020.

Li, X., Ling, F., Foody, G. M., Boyd, D. S., Jiang, L., Zhang, Y., Zhou, P., Wang, Y., Chen, R., and Du, Y.: Monitoring high spatiotemporal water dynamics by fusing MODIS, Landsat, water occurrence data and DEM, Remote Sens. Environ, 265, 112680, https://doi.org/10.1016/j.rse.2021.112680, 2021.

Line 358 - The text should describe how p was derived from the measurements.

Authors' response: Thank you for this comment. We added the statement to explain the calculation process of slope p values: "The actual bathymetric information obtained from field measurements is used to construct area-depth pairs. Subsequently, the slope profile p of the wetland depression is calculated based on the calculation formula established by Hayashi and Van der Kamp (2000)."

Line 362 - The text should define these methods.

Authors' response: Thank you for the suggestion. We added the following statement to illustrate the spatial prediction and modeling methods: "The spatial prediction and modeling methods assumes that the underwater slope profile is fundamentally similar to the above-water slope profile in wetland depressions."

Line 369 - This should be described in the methods section.

Authors' response: Thank you for the suggestion. We added the statement in methods section as "In general, DEM errors can be categorized into two types: systematic and random errors. To mitigate data noise, it is common practice to smooth the DEM before applying it for terrain analysis. Several filters commonly used for smoothing DEMs include median and mean filters, Gaussian filter, adaptive filter, and K-Nearest mean filter (Lindsay, 2016). In this study, we use the smoothed SRTM DEM derived from Gaussian filter to calculate the above-water slope profile."

Line 376 - This statement is not supported by the text - performance statistics for other methods are not provided.

Authors' response: Thank you for the comment. We rewritten the sentence to provide the performance metrics of all methods: "In summary, the comparative analysis reveals that the WetlandSCB framework demonstrates superior performance in bathymetric estimation for wetland depressional areas. For Baihe Lake, the slope profile p-value derived from the WetlandSCB framework (7.45) exhibits closer agreement with the actual measured value (7.29) than those obtained from the spatial prediction and modeling method (8.65) and its enhanced version incorporating smoothed SRTM DEM (8.51). Similarly, for Chagan Lake, the WetlandSCB framework yields a slope profile p-value (4.08) that more accurately approximates the actual value (4.05) compared to both the conventional spatial prediction and modeling method (4.78) and its enhanced version (4.37). These comparative results demonstrate the improved accuracy and reliability of the WetlandSCB framework in bathymetric characterization of wetland depressional areas relative to the other methods."

Line 382 - As part of the discussion, the text should compare the performance of the WetlandSCB in estimating wetlands depression storage capacity to the performance seen in simular studies.

Authors' response: Thank you for this suggestion. We added the following content to compare the performance of the WetlandSCB framework and other studies in estimating wetland depression storage capacity: "The use of elevation (to compute wetland depression

depths) and areal extent has emerged as an efficient method to estimate surface-water storage volume (Gao, 2015). After identifying wetland depressions, previous studies estimated the area and volume of each depression based on a statistical analysis of the DEM cells comprising that wetland depression (Rajib et al., 2018; Wu et al., 2019; Wu and Lane, 2016). This study compared and analyzed the water storage capacity of Baihe Lake and Chagan Lake, calculated using three medium-resolution 30-m DEM datasets: SRTM DEM, ALOS DEM, and MERIT DEM (Figure 13c). The results show that the accuracy of WDWSC calculation is highly dependent on the DEM data quality, with the MERIT DEM providing the most accurate results, with relative errors averaging 25.7% compared to the actual WDWSC. In contrast, the WDWSC calculation based on the WetlandSCB framework had relative errors of less than 10%, demonstrating that the WetlandSCB framework has the ability to accurately estimate WDWSC."

Gao, H.: Satellite remote sensing of large lakes and reservoirs: From elevation and area to storage, Wiley Interdiscip. Rev.-Water, 2(2), 147-157, https://doi.org/10.1002/wat2.1065, 2015.

Figure 13c. The calculation results of WDWSC based on three DEM datasets in validation wetland sites.

Line 390 - The text should describe how depression storage capacity was derived from the field-based hypersometric curves. The above water portion of the Chagan Lake curve is concave, indicating that the top of bank was not reached. This implies that the curve underestimates depression storage capacity.

Authors' response: Thank you for this comment. The process of calculating wetland depression storage capacity based on field-based hypsometric curves has been supplemented in the previous section. For Chagan Lake, its main area is represented by number 1 (below figure). Due to the influence of water level rise and artificial regulation, the areas marked as numbers 2, 3, and 4 can be connected to the main area. Therefore, only the water storage capacity within the main area was calculated.

Line 399 - The cylinders can be removed from the figures. They appear identical and therefore do not improve interpretation of the results.

*Authors' response*: Thank you for this comment. The cylinders in Figures 13a and 13b have been removed.

Line 424 - The conclusion section should include quantitative evidence of the method's performance.

Authors' response: Thank you for this comment. We added the following content in conclusion section:

(1) Processing by the morphological operators and prior information on water distribution map can accurately delineate the spatial extent of wetland depressions. The derived wetland depression map shows high spatial agreement with the true wetland depression map, achieving an overall accuracy and kappa coefficient both exceeding 0.95. The performance of the WetlandSCB framework is superior to the priority-flood algorithm in wetland

depression spatial delineation.

- (2) The water occurrence map can effectively correct numerical biases in above-water topography. Compared to original results, the corrected topography exhibits high consistency with true above-water topography, with average increases of 0.33 and 0.38 in Pearson coefficient and R2, respectively.
- (3) The coupling of spatial prediction and modeling methods with remote sensing techniques achieves high-precision estimation of underwater bathymetry of wetland depressions, demonstrating relative errors below 3% when compared to field measurements. The results prove that the superiority of the WetlandSCB framework over spatial prediction and modeling methods in underwater bathymetry estimation.
- (4) The WetlandSCB framework accurately estimates WDWSC with relative errors less than 10% compared to calculations based on field topography and bathymetry. This indicates that the WetlandSCB framework is suitable for accurate estimation of WDWSC in global-scale or areas without field measurement data.

Line 461 - These references are missing the year of publication.

**Authors' response**: Thank you for your careful reading. The year of publication has been placed after the DOI number in accordance with the HESS journal format.

Again, we sincerely thank the reviewer for their detailed and constructive comments and suggestions, which have helped us improve the quality and readability of our manuscript. Their invaluable time given for this review is greatly appreciate.

---

## Author Response (AR2)

**Reviewers' Comments and Authors Response**

**Comments from Editor**: One of the original reviewers has some minor comments. Please address those after which manuscript is ready to be published. Thanks.

Authors' response: We would like to express our sincere gratitude for your contribution to the review process and for your guidance throughout the revision of our manuscript. We also appreciate the constructive and insightful comments given by the reviewers, which have helped us further improve the quality of the work. We have carefully addressed the remaining minor comments and provided a point-by-point reply to the reviewer's suggestions. We are grateful for the opportunity to revise our manuscript and hope that the revised version meets the standards for publication in Hydrology and Earth System Sciences.

**Reviewer:**

1. Line 28 - typo: "filed".

Authors' response: Thank you for this comment. The word "filed" has been revised to "field".

2. Line 250 - "3-sigma rule" should be defined.

Authors' response: Thank you for this valuable suggestion. We have added a definition of the 3-sigma rule in the revised manuscript. The sentence has been revised as follows: "The 3-sigma rule is used to identify outliers in the underwater area-level pairs, defining an area-level pair as an outlier if its least squares residual exceeds three times the standard deviation."

Once again, we sincerely thank the reviewer for the constructive comments and suggestions, which significantly helped us improve the quality of our manuscript. We greatly appreciate the invaluable time dedicated to reviewing our work.